# Single molecule demonstration of Debye–Stokes–Einstein breakdown in polystyrene near the glass transition temperature

Nicole L. Mandel[1], Soohyun Lee[2], Kimyung Kim[2], Keewook Paeng [2✉] & Laura J. Kaufman [1✉]

Rotational-translational decoupling, in which translational motion is apparently enhanced over rotational motion in violation of Stokes-Einstein (SE) and Debye-Stokes-Einstein (DSE) predictions, has been observed in materials near their glass transition temperatures ($T_g$). This has been posited to result from ensemble averaging in the context of dynamic heterogeneity. In this work, ensemble and single molecule experiments are performed in parallel on a fluorescent probe in high molecular weight polystyrene near its $T_g$. Ensemble results show decoupling onset at approximately $1.15T_g$, increasing to over three orders of magnitude at $T_g$. Single molecule measurements also show a high degree of decoupling, with typical molecules at $T_g$ showing translational diffusion coefficients nearly 400 times higher than expected from SE/DSE predictions. At the single molecule level, higher degree of breakdown is associated with particularly mobile molecules and anisotropic trajectories, providing support for anomalous diffusion as a critical driver of rotational-translational decoupling and SE/DSE breakdown.

[1] Department of Chemistry, Columbia University, New York, NY, USA. [2] Department of Chemistry, Sungkyunkwan University, Suwon, Republic of Korea.
✉email: paeng@skku.edu; kaufman@chem.columbia.edu

Glass-forming materials have been the subject of extensive research, both for their utility in an array of technologies as well as for their unusual properties, including the apparent spatially heterogeneous dynamics in such systems that emerge near the glass transition temperature ($T_g$)[1–3]. Such dynamic heterogeneity manifests as stretched exponential relaxations in measured correlation functions in ensemble experiments, suggesting the presence of a wide array of underlying timescales. Non-exponential decays are also found in measurements of individual probe molecules in such systems, indicating single molecules can experience and report the full breadth of dynamics in small molecule and polymeric glass formers near their glass transition temperatures[4–6].

Dynamic heterogeneity in glass formers is believed to be causally related to other unusual phenomena observed in such systems, including the relationship between rotational and translational motion in violation of Stokes–Einstein (SE) and Debye–Stokes–Einstein (DSE) predictions, a set of predictions for translational and rotational dynamics originally derived for a large spherical solute in a hydrodynamic continuum[1,7–10]. While the SE and DSE relationships were originally derived for a large tracer particle, SE and DSE behavior have been found to hold in additional contexts including in liquids in which either small molecule tracers or self-diffusion was measured[11–15]. In contrast, SE and DSE predictions have often been found to fail in systems near their glass transition temperatures.

Specifically, the SE and DSE equations are given by $D_T = \frac{kT}{6\pi\eta r_s}$ and $D_r = \frac{kT}{8\pi\eta r_s^3}$, respectively, where $D_T$ and $D_r$ are the translational and rotational diffusion coefficients, respectively, $T$ is the temperature, $r_s$ is the hydrodynamic radius of a tracer particle, and $\eta$ is the host viscosity. Experimentally, rotational dynamics and DSE behavior are most commonly quantified through rotational relaxation time, $\tau_c$, with $\tau_c = \frac{4\pi\eta r_s^3}{3kT}$, and it has been noted that $D_r$ is not well-suited to investigating DSE behavior in supercooled liquids[16]. While the SE and DSE relationships predict that the translational diffusion coefficient, $D_T$, and the inverse rotational correlation time, $\tau_c^{-1}$, will both depend linearly on T/η, deviations from this prediction have been found experimentally in small molecule, polymeric, and colloidal systems near $T_g$[11,17–25] as well as in simulation[12,26–32]. The degree of deviation from the SE and DSE predictions varies between systems and generally increases upon cooling toward $T_g$. In most experiments on and simulations of molecular systems, translational motion has been shown to exhibit a weaker temperature dependence than rotation, resulting in apparent translational enhancement relative to rotation, up to 5 orders of magnitude at $T_g$[11,12,16–30,32]. Experiments showing this behavior have spanned modalities, some of which use probes and some of which measure self-diffusion. The particular deviation from SE and DSE behaviors linked to the apparent enhancement of translational dynamics relative to rotational dynamics in systems near $T_g$ has come to be known as rotational–translational decoupling.

Multiple explanations for rotational–translational decoupling and its temperature dependence in supercooled liquids have been proposed. Some hypotheses emphasize the presence of motion characterized by large jumps and unusually mobile subsets of molecules[12,27,29,30]. Related hypotheses emphasize persistence and exchange time contributions or cage rattling and breaking contributions, which can lead to non-Gaussian displacement distributions[22,28,33–38]. Many explanations for rotational–translation decoupling rest on arguments that it is a natural result of increasing dynamic heterogeneity with decreasing temperature near $T_g$[1,18–20,24,26,39–42]. In particular, it has been suggested that typical experiments that interrogate rotation preferentially weight slower sub-ensembles of molecules due to the long-time tail of rotational correlation functions, while typical experiments that probe translation preferentially weight faster sub-ensembles. Consistent with this explanation, individual molecules may not experience breakdown, but averaging over the behavior of many molecules will lead to the observed rotational–translational decoupling. Single molecule experiments, free of ensemble averaging, are uniquely primed to investigate the role of ensemble averaging in rotational–translational decoupling.

Extensive single molecule experiments characterizing rotational motion in supercooled liquids have been carried out[6,43–45]. While studies of rotations in supercooled liquids at the single molecule level have become commonplace, measurements characterizing translational mobility remain rare[46,47]. Until now, no experiment has simultaneously probed rotational and translational diffusion of single molecules in a supercooled host to characterize potential rotational–translational decoupling at the single molecule level.

In the current study, complementary ensemble and single molecule experiments that simultaneously measure rotational and translational motion of probes in a polymer film near $T_g$ are performed. Specifically, imaging fluorescence correlation microscopy (imFCM) and single molecule linear dichroism and localization microscopy are performed on N,N'-dipentyl-3,4,9,10-perylenedicarboximide (pPDI) fluorescent probe molecules in 168 kg/mol polystyrene. We find evidence of significant rotational–translational decoupling and SE/DSE breakdown in both imFCM and single molecule measurements, revealing that these phenomena persist at the single molecule level. Single molecule measurements allow stratification of molecules as a function of mobility, and we find a highly mobile sub-ensemble fully recapitulates the rotational–translational decoupling seen in the ensemble measurement. Additional insights obtained from single molecule trajectories suggest anisotropic translational mobility is associated with a higher degree of rotational–translational decoupling.

## Results

**Single molecule and ensemble measurements.** Two-channel wide-field single molecule movies were collected and analyzed for rotational and translational mobility at four temperatures near the glass transition temperature of high molecular weight polystyrene as well as at one temperature far below $T_g$, as described in "Methods". These data were directly compared to ensemble imFCM data using the same probe and host polymer.

Rotational analysis of single molecule data was performed on trajectories sufficiently long to accurately report the full breadth of heterogeneity in the system, as reflected by the median stretching exponent β value obtained from stretched exponential fits to single molecule linear dichroism autocorrelation functions ($C(\tau) = C(0) \cdot \exp\left(\tau/\tau_{fit}\right)^\beta$; $\tau_c = \left(\tau_{fit}/\beta\right) \cdot \Gamma(1/\beta)$); see "Methods" for additional details)[4,48]. We note that the decay of the linear dichroism autocorrelation is dominated by the second spherical harmonic[49–51]. The temperature dependence of median single molecule rotational correlation times, $\tau_c$, was found to be consistent with previous measurements in high molecular weight polystyrene[6,52]. Moreover, as expected based on previous single molecule results on pPDI in polystyrene, evidence of significant heterogeneity was found, with wide distributions of rotational correlation times (median FWHM = 0.96) and sub-unity values of the stretching exponent β (median β = 0.65)[6,43,44] (Fig. 1a, Supplementary 1).

For translational analysis, first, movies collected at 300.0 K, a temperature at which molecular motion was expected to be far below the noise floor, were analyzed to assess localization error in

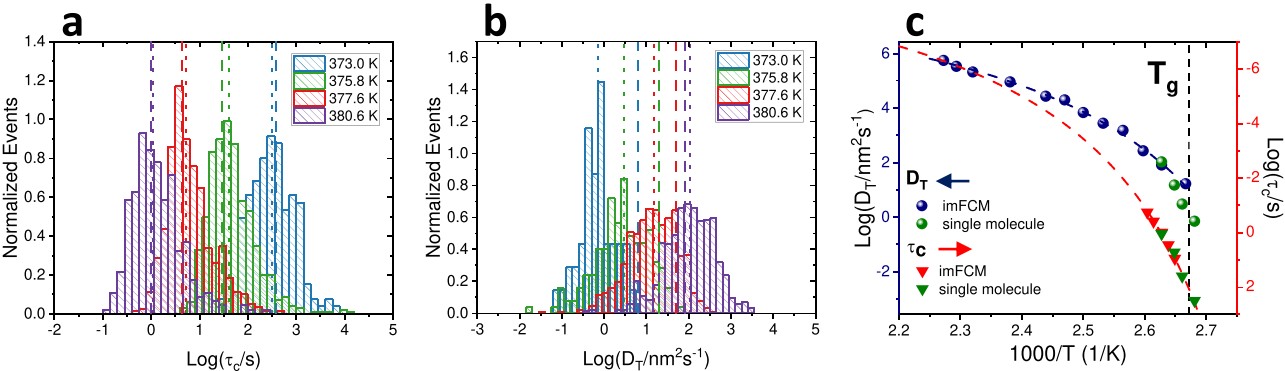

**Fig. 1 Comparison of single molecule and ensemble results. a** Single molecule rotational correlation time ($\tau_c$) distributions at four temperatures. Vertical dotted lines indicate median values and vertical dashed lines indicate $\tau_c$ values obtained from ensemble imFCM at the same temperature or as extrapolated from a best fit to the imFCM measurements (red dashed fitted line in (**c**)). In cases where a single vertical line is visible, the values are nearly identical. **b** Single molecule translational diffusion coefficient ($D_T$) distributions; vertical dotted lines indicate median values and vertical dashed lines indicate $D_T$ values at those temperatures or as extrapolated from a best fit to the imFCM measurements (blue dashed line in (**c**)). In (**a**) and (**b**) histograms are normalized by area under the curve. **c** Comparison of median single molecule results (green) to imFCM results in rotation and translation (red and navy, respectively). Navy dashed line is a polynomial fit to the data. Red dashed line is a Vogel-Fulcher-Tammann [$\log(\tau/\tau_0) = \frac{B}{(T-T_0)}$] fit to dielectric data[6,52], with fit values $\log(\tau_0) = -11.25$, $B = 414.15$ K, and $T_0 = 341.3$ K, vertically shifted by 0.75. Rotational and translational results overlap at high temperature, where no breakdown is expected, and the widening gap between rotation and translation as $T_g$ is approached from above indicates rotational–translational decoupling, with more apparent decoupling in imFCM compared to in median single molecule results. $T_g$, as indicated by a black dashed line, is determined by differential scanning calorimetry.

the single molecule translational mobility analysis. Localization error for data collected in a manner consistent with higher temperature measurements was found to be ≈10 nm, in good agreement with theoretical predictions (Supplementary 2). Empirically, the noise floor for translational diffusion coefficients was found to be ≈0.1 nm² s⁻¹. Subsequently, the same single molecule movies collected at 373.0, 375.8, 377.6, and 380.6 K and used to obtain rotational correlation times were merged as described in "Methods" and analyzed to obtain single molecule translational diffusion coefficients ("Methods" and Supplementary 3). Distributions of translational diffusion coefficients, $D_T$, were also found to be wide, with median FWHM of 1.37 (Fig. 1b), with no clear trend in width as a function of temperature.

In addition to assessing single molecule results as described above, quasi-ensemble (QE) reconstruction can also be performed with single molecule data. When this reconstruction was performed, $\tau_c$ values were nearly identical to median single molecule results while $D_T$ values were somewhat higher (Supplementary 4 and 6). Though quasi-ensemble reconstruction is meant to approximate ensemble results from single molecule data, it does not necessarily interrogate the full ensemble or the same ensemble interrogated by native ensemble approaches. As such, we also compared single molecule results to those obtained from imFCM measurements for the same probe/host pair over a temperature range including that probed in the single molecule study and extending to higher temperatures. Rotational correlation times and translational diffusion coefficients obtained from imFCM are shown as dashed vertical lines in Fig. 1a, b. Exemplary imFCM data is shown in Supplementary Fig. 5. While median values of rotational correlation times obtained from single molecule experiments and imFCM match, $D_T$ values show notable differences, with a narrower range of $D_T$ values obtained from the ensemble measurement relative to the single molecule measurements over the same temperature range. The largest discrepancy is found at the lowest temperature probed, with single molecule measurements showing a median value more than an order of magnitude slower than that obtained via imFCM measurements (Fig. 1b, c).

The median single molecule values of rotational relaxation times and translational diffusion coefficients in comparison to those

obtained from imFCM are shown in Fig. 1c. Recalling the SE and DSE equations, $D_T = \frac{kT}{6\pi\eta r_s}$ and $\tau_c = \frac{4\pi\eta r_s^3}{3kT}$, we note $D_T\tau_c = \frac{2}{9}r_s^2$, a relationship that itself is considered an expression of DSE behavior[8,10,19]. This expression of DSE behavior is particularly useful given that viscosity can be challenging to measure experimentally and attain from simulation data[53,54]. Translational data obtained from imFCM and Volgel–Fulcher–Tammann extrapolation of rotational data obtained from imFCM converge at the three highest temperatures probed, to a value $D_T\tau_c ≈ 0.25$ nm², yielding a physically reasonable pPDI radius estimate of 1.06 nm. With decreasing temperature, divergence in $\tau_c$ and $D_T$ values emerges and increases as temperature decreases in both imFCM and single molecule experiments, revealing a large degree of rotational–translational decoupling, similar to what has been described previously in polystyrene and greater than that found in most other measured systems[20]. While small differences in the temperature dependence of $\tau_c$ have been shown to exist depending on the spherical harmonics interrogated[49–51], such findings are inconsistent with the very large degree of rotational–translational decoupling seen in this study.

While rotational results from both single molecule and imFCM measurements exhibit very similar values and temperature dependences, translational results vary by method. As $T_g$ is approached from above, the difference between the translational diffusion coefficients obtained from imFCM and those obtained from single molecule measurements widens, with single molecule translational data exhibiting slower translation near $T_g$ relative to the imFCM result. The ensemble imFCM measurements show initial decoupling at ≈1.15$T_g$, with onset of decoupling determined as the point at which the extrapolated curve deviates more than 0.1 decades from the DSE prediction. Decoupling then increases as temperature decreases, with breakdown of over 3.5 orders of magnitude at $T_g$.

Single molecule results also show evidence of rotational–translational decoupling across the full temperature range interrogated, with the degree of translational enhancement increasing as $T_g$ is approached. While significant decoupling is seen across both types of measurements, particularly at the lowest temperatures interrogated, median

single molecule results show less translational enhancement than do imFCM measurements (≈2.5 vs. 3.5 decades), suggesting one decade of the rotational–translational decoupling observed in imFCM emerges from ensemble averaging at these temperatures. Supplementary Information 6 shows information also shown in Fig. 1c with additional detail on the full distribution of single molecule results and data points obtained from QE reconstruction of single molecule rotational and translational results. We note that degree of breakdown is greater for quasi-ensemble calculated values than median values from single molecule results, suggesting a degree of the "excess" breakdown seen in imFCM is attributable to ensemble averaging over the molecules assessed in the single molecule measurements. Additional discussion of median single molecule results in the context of SE/DSE breakdown is presented in Supplementary 7.

**Individual single molecules show significant rotational–translational decoupling.** While comparing behavior obtained from ensemble and single molecule measurements provides some insight into the nature of rotational–translational decoupling, single molecule measurements also allow investigation of individual molecules' rotational and translational dynamics assessed simultaneously. This can then reveal, for example, whether an individual molecule shows enhancement in its translational diffusion relative to its rotational dynamics over the same period. Figure 2 shows data from individual molecules presented in scatter plots, with each point representing rotational and translational data obtained from a single molecule. Single molecule $\tau_c$ vs. $D_T$ values are shown in Fig. 2a, with the black line indicating the expected result in the absence of rotational–translational decoupling. The overall temperature dependence is consistent with expectation from DSE behavior, as is seen in the similarity in slope between the single molecule data and the line associated with DSE behavior. However, at each temperature considered, there is limited correlation between individual molecule $D_T$ and $\tau_c$ values. This finding may be related to the fact that translational mobility typically could only be tracked for a subset of the full trajectory analyzed to obtain $\tau_c$ due to out-of-plane rotation that inhibited molecule tracking. However, all attempts to mitigate differences in translational and rotational assessment led to similar results (Supplementary 8). As also shown in Fig. 1c, rotational–translational decoupling is seen at every temperature, with $D_T$ higher than expected for nearly all molecules for the $\tau_c$ obtained from the same molecule. Moreover, individual molecules clearly show different degrees of breakdown, with a few molecules showing negligible breakdown and others showing up to an order of magnitude greater breakdown than the median molecule at a given temperature. Figure 2b shows the relationship between $D_T$ and degree of DSE breakdown as captured by the deviation of $D_T\tau_c$ from the expected value of 0.25 nm². The strong positive correlation at each temperature suggests that DSE breakdown is driven primarily by translation, with individual molecules with higher diffusion coefficients exhibiting larger degrees of breakdown.

Focusing on particularities of individual molecules' translational trajectories, Fig. 2c, d shows relationships between deviation from DSE behavior and radius of gyration and trajectory anisotropy, respectively. Molecules with a larger radius of gyration exhibit greater degree of DSE breakdown (Fig. 2c), consistent with the idea that motion through distinct dynamic domains enhances breakdown. Moreover, molecules exhibiting high degrees of breakdown tend to have more anisotropic trajectories (Fig. 2d), suggesting these molecules may exhibit directionally persistent motion, reminiscent of predictions of facilitated models[34–37].

In addition to characterizing single molecules as described above, following Flier[46], single molecule trajectories were categorized as mobile or immobile based on the radius of gyration compared to that measured at 300.0 K, well below $T_g$, where measured motion can be ascribed to noise (Supplementary 2). Using this method, molecules with time-normalized radii of gyration greater than 95% of molecules at 300.0 K were classified as mobile, and the remainder were classified as immobile. In practice, molecules with $R_g > 16.9$ nm were identified as mobile (Fig. 3a). We note that this approach results in more molecules analyzed than does the approach in which molecules are assessed following mean square displacement analysis, as in such cases the least mobile molecules may yield non-physical (negative) diffusion coefficients and such molecules are not considered in subsequent analysis (Supplementary 9 and 10). The percentage of molecules deemed mobile increased with temperature, up to nearly 100% at the highest temperature investigated (Fig. 3b). When assessed separately, mobile molecules were found to have higher diffusion coefficients (Fig. 3c, Supplementary 9, 10), more anisotropic trajectories (Fig. 3d), and a greater degree of DSE breakdown (Fig. 3e) than immobile molecules. In particular, for mobile molecules, the degree of DSE breakdown increases more abruptly as temperature decreases relative to the results when all molecules are considered, matching that of ensemble imFCM (Fig. 3e). For all but the highest temperature, mobile molecules also tended to have lower β values, suggesting these molecules experience a wider range of dynamic environments than molecules deemed immobile (Supplementary 10). The fact that the temperature dependence of DSE breakdown exhibited by mobile molecules matches that seen in ensemble imFCM measurements is consistent with the idea that more mobile molecules with more anisotropic trajectories drive rotational–translational decoupling and DSE breakdown[27].

## Discussion

Rotational–translational decoupling in violation of DSE predictions has long been found in materials near their glass transition temperature and has generally been thought to be the result of averaging over molecules and dynamically heterogeneous regions in these materials. Single molecule studies eliminate the effects of averaging over molecules and can thus reveal whether and the degree to which rotational-translational decoupling and DSE breakdown are single molecule phenomena. The present results, for the first time, simultaneously analyze single molecule rotation and translation of probe molecules in a glass former near $T_g$, measurements performed in parallel with ensemble studies of the same probe/host system. These measurements show that rotational–translational breakdown in violation of DSE predictions persists at the single molecule level and cannot be purely the result of averaging over distinct sub-ensembles of slow and fast molecules. However, given that molecules must be observed for finite times to characterize their rotational and translational dynamics, time averaging and spatial averaging over potential dynamic heterogeneity does exist in single molecule data. The manifestations of such averaging have been seen in both simulations and experiments focused on rotations of single molecules in systems near $T_g$, where the value of the stretching exponent β and its change as a function of observation time are sensitive measures of spatiotemporal averaging over distinct dynamic environments in the sample[6,43,44,48,55].

We interpret the results of the measurements here, broadly, as an indication that averaging over distinct dynamic domains is responsible for much of the observed rotational–translational decoupling, with the relative spatial (and ensemble) averaging decreasing from the imFCM measurements to the single molecule

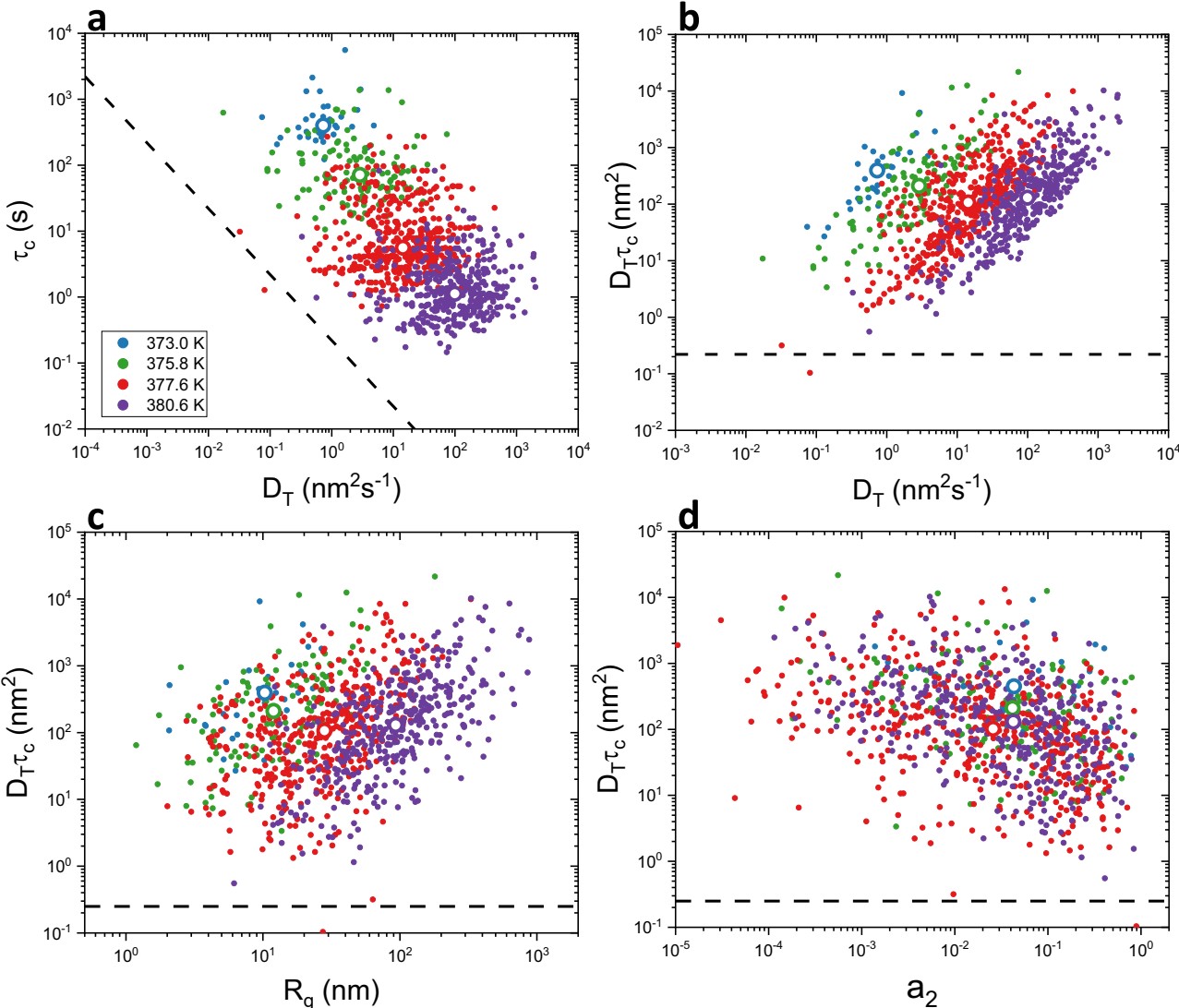

**Fig. 2 Scatter plots of rotational and translational data obtained from single molecule trajectories.** Each point represents a single molecule. Large open points show median single molecule values. Black dashed lines in all panels represent $D_T\tau_c = 0.25$ nm$^2$, the expected value in the absence of DSE breakdown. **a** Rotational correlation time ($\tau_c$) vs. translational diffusion coefficient ($D_T$). **b** Value of $D_T\tau_c$ vs. $D_T$. Strong correlation between degree of breakdown as captured by deviation of the value of $D_T\tau_c$ from that expected (black dashed line) and $D_T$ suggests breakdown is primarily driven by translation. **c** Radius of gyration ($R_g$) vs. the value of $D_T\tau_c$ shows molecules that explore larger areas tend to exhibit higher degree of breakdown. **d** Asymmetry coefficient ($a_2$) vs. the value of $D_T\tau_c$, with $a_2 = 1$ indicating an isotropic trajectory and smaller values indicating increasingly directional trajectories. Molecules with anisotropic trajectories tend to exhibit more breakdown. In all panels, color coding for temperature is as shown in the legend of (**a**).

measurements. In particular, for single molecule measurements, we estimate the average spatial scale over which averaging takes place from the most probable step size associated with these measurements over the lag time used to characterize the translational diffusion coefficient, i.e. the first six lag times of the MSD (Supplementary Fig. 4b). This length scale is 70–100 nm at all temperatures and corresponds to a timescale of $\approx 3\tau_c$. These results suggest that most, but not all, rotational–translational decoupling observed emerges from molecular behavior over length scales shorter than 100 nm. As more mobile molecules traverse such length scales more quickly, the fact that mobile molecules fully recapitulate ensemble breakdown also supports this conclusion. Additionally, the high degree of correlation between $D_T$ values, anisotropy of translational trajectories, and deviation from DSE behavior, supports previously posited explanations that anomalous diffusion in the context of dynamic heterogeneity contributes to violation of DSE predictions[27,29,30].

This is further supported by the fact that step size distributions of molecules at temperatures above $T_g$ exhibit long, non-Gaussian tails (Supplementary 11).

In sum, simultaneous measurement of rotational and translational motion of single fluorescent probe molecules in high molecular weight polystyrene near its glass transition temperature reveal that very significant deviations from DSE behavior persist at the single molecule level; thus, DSE breakdown and rotational–translational decoupling cannot emerge solely from preferential weighting of molecules with slow or fast dynamics, as may occur in ensemble rotational and translational experiments, respectively. Our results further suggest that length scales of $\approx 100$ nm and timescales of $\approx 3\tau_c$ are sufficient to capture most breakdown seen at the ensemble level. The degree of breakdown is determined primarily by enhanced translation and is driven by particularly mobile molecules. When mobile molecules are considered in isolation, their (median) behavior fully recapitulates

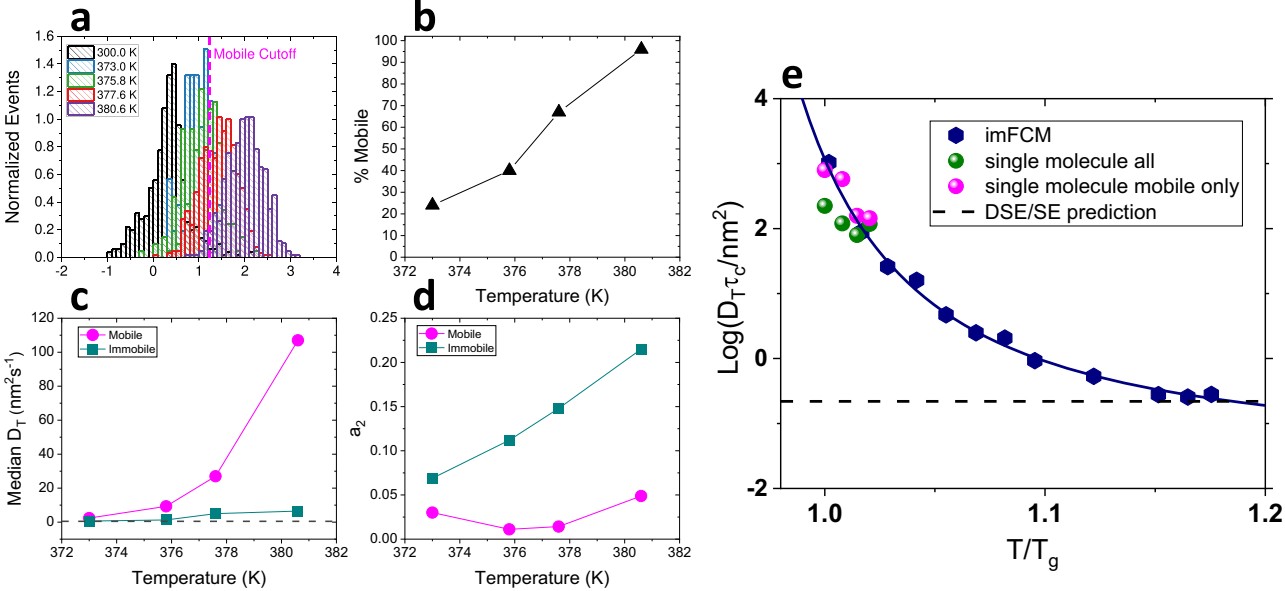

**Fig. 3 Analysis of mobile and immobile single molecule trajectories. (a)** Distributions of time-normalized radii of gyration. Mobile cutoff was determined to be 16.9 nm, as indicated by the pink dashed line. **(b)** Percentage of molecules identified as mobile increases as a function of temperature. **(c)** Median diffusion coefficients for mobile and immobile molecules as a function of temperature. Dashed line indicates median $D_T$ from 300.0 K baseline data with only positive diffusion coefficients included. **(d)** Median asymmetry coefficient ($a_2$) as a function of temperature for molecules identified as mobile and immobile. Values closer to 0 indicate more anisotropic trajectories. **(e)** Degree of DSE breakdown vs. temperature as obtained from imFCM (navy symbols) and median single molecule results for all (green) and mobile (pink) molecules. Navy line is a polynomial fit to the imFCM data. Black dashed line indicates DSE behavior.

ensemble levels of breakdown, though individual molecules show deviation from DSE behavior that spans more than an order of magnitude. Finally, single molecule deviation from DSE behavior is seen most readily in molecules displaying anisotropic trajectories, providing support for explanations of DSE breakdown that focus on anomalous diffusion as may occur in particular types of cooperative and facilitated motion[27,29,30,34–38].

## Methods

**Sample preparation**. Polystyrene (atactic, $M_w$ = 168 kg/mol, Đ = 1.05) (PS) was purchased from Polymer Source, re-precipitated in hexane 4x, and dissolved in toluene at a concentration of 3.5–4.0 wt% polystyrene. The resulting solution was photobleached in a home-built high-power LED setup for 72 h to ensure a non-fluorescent host. The fluorescent dye N,N'-dipentyl-3,4,9,10-perylenedicarbox-imide (pPDI) was purchased from Sigma Aldrich and dissolved in toluene, then mixed with the polymer solution, resulting in a probe concentration of ≈$10^{-11}$ M for single molecule experiments or ≈$10^{-8}$ M for imFCM experiments. Silicon wafers were cut to ≈7 × 7 mm squares and cleaned with piranha solution ($H_2SO_4$:$H_2O_2$ = 3:1). The pPDI in PS solution was spin coat (Laurell, WS-650Mz-23NPPB) onto a cleaned silicon wafer at 2000 RPM. The spincoat film was measured to be ≈200 nm thick by ellipsometry (Nanoview, SE MG1000), sufficiently thick such that measurements are expected to be dominated by bulk dynamics[56]. After spin coating, the concentration of pPDI was ≈$10^{-10}$ M for single molecule experiments and ≈$10^{-7}$ M for imFCM experiments. This resulted in ≈150–200 features per field of view for single molecule experiments and 50–100 probes per volume represented by a pixel in imFCM experiments. In the single molecule measurements, this concentration is sufficiently low such that diffraction-limited fluorescence from individual molecules will not typically overlap, and trajectories of mobile molecules do not typically intersect over the course of the measurement. All measurements were performed under vacuum at ≈20 mTorr after annealing in the vacuum chamber for at least 12 h at $T_g$ + 20 K to ensure full removal of toluene.

**Data collection**. Both single molecule and imFCM rotation and translation data were collected on home-built wide-field fluorescence microscope setups. A continuous wave laser (532 nm, CNI, MGL-III-532) was passed through a line filter (Semrock, LL01-532-12.5) to eliminate optical noise and coupled into a multimode fiber. The fiber was continuously shaken by a speaker during measurement to ensure a homogeneously illuminated field of view with random polarization. The light was focused onto the back of an objective lens (Olympus, M Plan Apochromat MPLAPON, 100x, NA = 0.95, WD = 0.3 mm) inside of a vacuum and temperature-controlled chamber, illuminating the sample. Excitation power was ≈15 mW at the back of the objective lens, corresponding to a power density of

≈210 W/cm² at the sample. Fluorescence was collected in the epi-direction through the objective lens, and passed through a dichroic mirror (Semrock, LPD02-532RU), a longpass filter (Semrock, LP03-532RE-25) and a bandpass filter (Semrock, FF01-571/72-25) to eliminate spectral noise and excitation light. The field of view was ≈85 μm in diameter, and 1 pixel corresponded to 169 ×169 nm² of real space, as confirmed with measurements using a Ronchi ruling. For single molecule measurements, emission was split into two orthogonally polarized channels (s and p) via a Wollaston prism and collected on an electron-multiplying charge-coupled device camera (EMCCD; Andor, iXon Ultra 897). The magnitude of s- and p-polarized excitation intensities was fine-controlled to be equal at the sample by a half and quarter waveplate placed between the laser and multimode fiber coupler.

The glass transition temperature of the sample was determined to be 374.3 K by differential scanning calorimetry (DSC, Q20a, TA Instruments). At least three scans were performed, and $T_g$ was determined as the midpoint of the averaged cooling scans after the first run. For single molecule measurements, data were collected at five temperatures: 300.0, 373.0, 375.8, 377.6, and 380.6 K. Temperature was determined by a thermocouple at the sample coupled with calibration based on comparison to previous temperature-dependent data on pPDI in 168 kg/ml PS, including correction for sample heating associated with continuous exposure[44].

Data were collected at 300.0 K ($T_g$ – 74.3 K) to establish a static error baseline in localization microscopy. These data were collected continuously at a frame rate of 0.5 Hz. Frame rates for the 4 other temperatures were chosen to correspond to ≈15–40 frames/rotation time ($\tau_{fit}$). In practice, data were collected at frames rates of 0.2, 0.5, 5, and 50 Hz for 373.0 K, 375.8 K, 377.6 K, and 380.6 K, respectively. Exposure time was 2.0 s at 373.0 and 375.8 K, 0.2 s at 377.6 K, and 0.02 s at 380.6 K. Illumination was shuttered between frames for movies collected at 373.0 K to limit photobleaching. The number of frames collected per movie was chosen to result in trajectory lengths of ≈250–300 $\tau_{fit}$, sufficiently long to avoid statistical effects associated with short trajectories for extraction of rotational correlation times and associated variables[4]. For imFCM, data were collected at 377.2, 379.0, 380.5, 382.5, and 384.0 K for rotation at frame rates that corresponded to 10–20 points/$\tau_{fit}$ (red triangles, main text Fig. 1C). For translation (navy circles, Fig. 1C), data were collected at 375.0 K (432), 380.6 (256), 385.0 (323), 389.9 (47), 394.9 (49), 400.0 (52), 404.9 (47), 410.0 (52), 420.0 (39), 431.0 (34), 440.0 (42), and 436.0 K (67); the numbers in parentheses are frame rates in terms of $\tau_D$, the time required for the ACF decay due to translation to drop to ½ of its initial value.

### Data analysis

*Wide-field single molecule measurements*. Rotational analysis on two-channel movies was performed via autocorrelations of linear dichroism, as described previously[57]. Movies were analyzed using Interactive Data Language (IDL) software (ITT Visual Information Solutions). Features were chosen from a summed and bandpassed 500-frame section of the middle of each movie using the "feature" algorithm described in Ref. [58]. Subsequent analysis was performed on raw and

unfiltered images. Fluorescence intensities of each found molecule in two orthogonally polarized channels ($I_s$ and $I_p$) were recorded from each frame. From these intensities, single molecule linear dichroism (LD) was calculated for each frame via $LD(t) = (I_s(t) - I_p(t))/(I_s(t) + I_p(t))$, where t is chronological time. The full LD trajectory for each molecule was used to calculate an autocorrelation function (ACF) using $C(\tau) = <a(t) \cdot a(t + \tau)>/<a(t)>^2$, where $a(t) = LD(t) - <LD(t)>$. Each ACF was then fit to a stretched exponential function, $C(\tau) = C(0) \cdot \exp(\tau/\tau_{fit})^\beta$ using least-squares fitting until the function decayed to 0.1, where $\tau$ is lag time, $\tau_{fit}$ is a timescale characterizing the early portion of the decay that we term rotation time, and $\beta$ is the stretching exponent. Only molecules with $0.2 < \beta < 2$ and (number of frames)/$\tau_{fit}$ >2 were included in subsequent analysis to assure data was well fit and physically reasonable. The average rotational correlation time for each molecule, $\tau_c$, was calculated from the fit values of $\tau_{fit}$ and $\beta$ via $\tau_c = (\tau_{fit}/\beta) \cdot \Gamma(1/\beta)$, where $\Gamma$ is the gamma function. Quasi-ensemble analysis was performed by adding all single molecule ACFs from a single movie and fitting the resulting decay to the stretched exponential equation above.

In advance of translational analysis, two-channel movies were combined into a single channel. Channels were cropped, aligned and summed. For movies collected with exposure times less than 2 s (377.6 K and 380.6 K), frames were summed to achieve effective exposure times of 2 s for 377.6 K and 1 s for 380.6 K. While movies were collected at 15-40 frames per $\tau_{fit}$ at each temperature, intermediate frame deletion was performed, and data was analyzed at 1 frame/$\tau_{fit}$ except where noted. This approach provided constant average rotational motion between assessed positions across temperatures. Following intermediate deletion, trajectories of 100-300 frames (100–300 $\tau_{fit}$) taken around the temporal center of the full movie were analyzed. In particular, features were identified by intensity and tracked via the ImageJ plugin ParticleTracker, which uses iterative intensity-weighted centroid calculation to identify and greedy hill-climbing optimization with topological constraints to link detected particles between frames, as described in Sbalzarini and Koumoutsakos[59]. Trajectories longer than 20 frames were analyzed in Igor Pro software (WaveMetrics) via construction of mean-square displacements (MSD) in two dimensions from individual molecules' x and y coordinates, via $MSD = <|r - r_0|^2> = <(x(t) - x(0))^2 + (y(t) - y(0))^2>$, where t is lag time and $r_0$ is the reference position for the molecule. The first 6 points of MSD vs lag time were fit by linear regression, allowing a non-zero intercept consistent with the presence of localization error. Diffusion coefficients were calculated from the slope via $MSD = 4D_T t + \varepsilon$, where $\varepsilon \approx 4\sigma^2$, and $\sigma$ is localization error. Increasing the number of frames analyzed did not result in longer trajectories, as molecules could rarely be tracked beyond 50 frames before being lost due to intensity fluctuation from out-of-plane rotation. Quasi-ensemble MSDs were calculated by averaging all single molecule MSDs at the first 6 lag times and fitting resulting points to a line with slope $4D_T$. From the trajectories that yielded diffusion coefficients, radii of gyration ($R_g$) were obtained by calculating the radius of gyration tensor $\hat{T}$,

$$\hat{T} = \begin{bmatrix} \frac{1}{N}\sum\limits_{j=1}^{N}(x_j - \langle x \rangle)^2 & \frac{1}{N}\sum\limits_{j=1}^{N}(x_j - \langle x \rangle)(y_j - \langle y \rangle) \\ \frac{1}{N}\sum\limits_{j=1}^{N}(x_j - \langle x \rangle)(y_j - \langle y \rangle) & \frac{1}{N}\sum\limits_{j=1}^{N}(y_j - \langle y \rangle)^2 \end{bmatrix} \quad (1)$$

whose eigenvalues are $R_1$ and $R_2$. These eigenvalues were used to calculate time-normalized $R_g$ via $R_g = \sqrt{R_1^2 + R_2^2} \cdot \sqrt{t_{total}/t_{traj}}$, where $t_{total}$ is an arbitrarily chosen time for normalization (1000 s) and $t_{traj}$ is the length of the individual molecule's trajectory in seconds[46]. The eigenvalues were also used to calculate asymmetry coefficients ($a_2$) where $a_2 = R_2^2/R_1^2$, where $a_2$ values range from 0 to 1, with 1 indicating an isotropic trajectory[60].

Molecules found via rotation and translation analysis were matched by a Python program based on their (x,y) positions identified in rotational and translational analysis, assuming molecules found within 2 pixels to be the same molecule. As probe molecule concentration was low, this rarely resulted in mismatching.

**Imaging fluorescence correlation microscopy.** From collected movies, time-sequenced fluorescence intensities ($I(t)$) were extracted pixel by pixel from a 30×30 pixel region (50 × 50 μm²), resulting in 900 intensity traces. Autocorrelations of each trace were constructed via $G(\tau) = \langle \delta I(t) \cdot \delta I(t + \tau) \rangle / \langle (\delta I(t))^2 \rangle$, where $\delta I(t) = I(t) - \langle I(t) \rangle$. All autocorrelation decays obtained were then averaged to obtain a single autocorrelation decay. The extraction of intensities and the construction of the single autocorrelation decay were performed using the ImageJ Imaging FCS 1.52 plugin[61], which accounts for pixelation and overlap of the point spread function (PSF) of single point-like particles in an image detection setup[62]. The averaged autocorrelation decay was fit to Eq. (2), where the first part of the equation captures decay due to rotation and the second to translation[63]. Autocorrelation decay potentially caused by intersystem crossing was ignored, as the fluorescence quantum yield of the pPDI probe is close to unity ($\approx 0.97$)[63,64].

$$G(\tau) = \left[1 + C(0) \cdot e^{-\left(\frac{\tau}{\tau_{fit}}\right)^\beta}\right] \cdot \frac{1}{N} \left[\frac{\text{erf}(p(\tau)) + \frac{e^{-p(\tau)^2} - 1}{\sqrt{\pi}p(\tau)}}{\text{erf}(p(0)) + \frac{e^{-p(0)^2} - 1}{\sqrt{\pi}p(0)}}\right]^2 + G_\infty; p(\tau) = \frac{a}{\sqrt{4D_T\tau + w_{xy}^2}} \quad (2)$$

In Eq. (2), erf represents the error function, a is the length scale represented by a pixel in real space, and $w_{xy}$ is the calibrated lateral PSF, which is the $1/e^2$ radius of the Gaussian approximated PSF ($w_{xy} = \sigma_0\lambda_{em}/NA$), where $\sigma_0$ is a calibration factor determined to be 0.95 and $\lambda_{em}$ is 556 nm, which is the weight average of the fluorescence spectrum of pPDI that passed the fluorescence filters. Fitting parameters for the translational portion of the decay were N (number of diffusing probe molecules in an effective area defined by the convolution of the square area of the pixel ($a^2$) with the PSF)[65] and $D_T$ (translational diffusion coefficient), and for the rotational portion of the decay were $\tau_{fit}$ (rotation time) and $\beta$ (stretching exponent, as also described above). In practice, while accessing and fitting both rotation and translation portions of the autocorrelation is possible (Supplementary 5), rotational and translational data were collected and analyzed separately except where noted.

## Data availability
The datasets, including raw movies, generated during and/or analyzed during the current study are available from the corresponding authors on request.

## Code availability
All code used in analysis of data in the current study is available from the corresponding authors on reasonable request.

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

## Acknowledgements

L.J.K. acknowledges support from the National Science Foundation (CHE-1660392 and CHE-1954803). K.P. acknowledges support from the National Research Foundation of Korea (NRF) (NRF-2020R1A2C1013896). We thank Alec Meacham, Talha Rehman, Kelli Mandel, Alex Devanny, Mark Ediger, and Ludovic Berthier for discussions and/or assistance with data analysis and interpretation.

## Author contributions

L.J.K. and K.P. conceived the idea and directed the project. N.M. and S.L. collected single molecule data and N.M. analyzed single molecule data. K.K. collected and analyzed imFCM data. N.M., L.J.K., and K.P. wrote the manuscript. All authors commented on the manuscript.

## Competing interests

The authors declare no competing interests.
