## [Peer Review File · Nature Communications]

Single Molecule Demonstration of Debye-Stokes-Einstein
Breakdown in Polystyrene near the Glass Transition
TemperatureReviewers' Comments:

Reviewer #1:

Remarks to the Author:

The authors discuss translational and rotational molecular motions and their decoupling behavior of small solutes in polystyrene environments, and physical properties of the Stokes-Einstein-Debye relationship. The Stokes-Einstein (SE) and Stokes-Einstein-Debye (SED) relationship has been derived in hydrodynamic theorem on the basis of the Stokes equation, and their violations have been very highlighted in the large field of physical chemistry, applied physics, and materials chemistry. The authors employed experimental measurements of the fluorescent probe technique to quantify single-molecule and ensemble diffusions in polymer glass.

Their measured diffusion coefficients show the strong decoupling behavior of translational-rotational motions near the glass transition temperature, resulting the violation of SED relationship. The results of single-molecule and ensemble measurements agree well each other, and the single-molecule one is further applied to reveal structural effects of the gyration radius of diffusion solutes. I think that these experimental data help to understand nanoscale behavior and new physical insights on the SE and SED relationship in glassy polymer, and will attracts wide attentions of many readers in Nature Communications (in my opinion, Fig 3e is especially impressive). Thereby, I generally agree with the publication of this study from NatComm, but in my opinion, the following modifications should be required with careful consideration.

- in abstract: the authors said that translational diffusion coefficients nearly 400 times higher than expected from "SED", but I think that the translational diffusion should be related with the "SE", not SED. Why it's compared with the SED equations?

- in abstract and main body: I'm confusing the phrase of "anomalous diffusion". The non-linear relaxation of MSD is well discussed like that, for example, in JCP 143, 191103 (2015). However, in this study, the time relaxation of MSD seems to be approximated as $MSD \sim 4Dt^1$, which is different with the above article. What is the correct mean of "anomalous diffusion" the authors said? (I think that this phrase would be sometimes misleading)

- in the introduction of main body: The introduction of this manuscript is well written for the physical perspectives of dynamic heterogeneity, glass transition, and polymer chemistry. However, the present story remains unkind for theoretical backgrounds of the SE and SED relationships with the Debye relaxation model. A lot of experimental and computational studies have been reported until now for applicability and its violation of the hydrodynamic relationships, which should be discussed in the introduction, and compared with the present focus for many readers in the wide varieties of scientific fields.

- lines 39, 40: the SE and SED relation should be emphasized to be derived on the basis of hydrodynamics. In the present case, the condition of small-molecule solute also does not follow the hydrodynamic assumption. However, for example, the following references empirically supports the applicability of SE and SED relationship under low-viscous condition, not high-viscous conditions. These references should be mentioned in the introduction.

-- J. Chem. Phys. 126, 224506 (2007).

-- J. Phys. Condens. Matter 33, 055401 (2021).

- lines 43-45, page 1: the authors should clarify convincing citations for the sentence of "Most often, translational motion has been shown to exhibit a weaker temperature dependence than rotation,". Furthermore, what is the mean of "Most often"? It's apparently related to the phrase of "anomalous diffusion"...

- Fig 1: the bar plots of Fig.1, Fig.S1, Fig.3 etc is very unkind to identify the median and averaged value of single-molecule rotational correlation time from the Figures. The overlapped areas should be also shown for readers.

- Fig 2: If my understanding is correct, the product of D_T and τ_c is obtained from each state of single molecule observation. If so, the authors clarify the difference of notations for D_t and averaged (or median) D_T in the data of single molecule measurement; for example, Fig 2a uses the sampling set of (D_T^i, τ_c^i) and Fig 1c uses the averaged (or median) D_T and τ_c .

Furthermore, if it's possible, it's may be better to show the plots of Fig 2 as the contour map of probability density-of-states, because the most-probable state is hidden in the case of high-density scatter plot. Otherwise, please plot the median values of D_T and τ_c used in Fig 1c and discussed it.

- Fig 2c: the authors discussed the effects of gyration radius in the SED relation, and concluded that the larger gyration radius enhances the SED violation. Meanwhile, the correlation of Fig 2c appear to be weak. Please change it to the probability density-of-state as above, or quantify the Pearson's product moment correlation coefficient. Furthermore, in main body, it's better to discuss theoretical relationship with the equation of $D_T \tau_c$ for Fig 2c, as mentioned in Supporting Information.

- Throughout the manuscript, the authors discuss the behavior of diffusion coefficients at the 4 states of different temperatures, but the solvent effects of polystyrene is not considered. The temperature difference changes the free-volume of small solutes, which affects the gyration radius of movable solutes plotted in Fig 2c. I think that the authors should mention this solvent effects of polymer glasses on translational-rotational decoupling and the SED relationship.

Reviewer #2:

Remarks to the Author:

In this manuscript, Mandel et al. showed single molecule experimental results of Debye-Stokes-Einstein (DSE) Breakdown in polystyrene glasses. In particular, the authors demonstrated translational-rotational decoupling towards the glass transition temperature by measuring the temperature dependence of the product of diffusion coefficient D and rotational correlation time τ_c . Furthermore, the degree of the DSE breakdown is explained by the mobile molecule contributions from the single molecular level analysis.

Although I'd appreciate the report of many results using single molecule experiments, the result of DSE breakdown is not surprising and a great deal in the present manuscript has already been addressed in previous papers by experiments and simulations, as cited by the authors. The manuscript could be improved with more extensive discussions, particularly for what the new physical implication of DSE breakdown beyond current understanding is from single molecule analysis.

Below, I'd like to list several questions and comments.

(1) The definition of SE/DSE relationships should be discussed in a more strict manner. Indeed, there have been other presentations of SE/DSE relationships, including $\eta/(\tau_c * T)$ with viscosity η and D/D_r with rotational diffusion constant D_r , different from $D * \tau_c$. Why $D * \tau_c$ only? If those quantities would be assessed and compared consistently, that would be beyond the current understanding of SE/DSE breakdowns in glassy systems.

(2) In the Debye model, the rotational correlation depends on the order of spherical harmonics. It is important to discuss the order dependence of τ_c because the validation of the Debye model is the basis of DSE quantity, $D * \tau_c$.

(3) The term 'decoupling' is used many times in the manuscript, but it seems ambiguous. In fact, Figure 2(a) shows a correlation between D and τ_c , indicating that more diffusion leads to more rotation. It may be 'translational-rotational coupling', but would be wrong?

(4) The contribution of mobile molecules to DSE breakdown is interesting. It is also interesting to investigate the effects of immobile counterparts on DSE relationship. From Figure 2(b), it would be expected that $D \cdot \tau_c$ of the immobile subset will exhibit an almost recovery of DSE relation. If so and if not so, what is the implication of immobile molecules in DSE breakdowns?

(5) Another interesting quantify to highlight between mobile and immobile molecules is the ratio of $\langle \tau_c \rangle \langle 1/\tau_c \rangle$ or $\langle D \rangle \langle 1/D \rangle$. Is it possible to measure those values?

In summary, I would recommend that this manuscript is not suitable for publication in Nature Communications, but be publishable in more specialized papers with lower requirements of novelty after adequate revisions.

Reviewer #3:

Remarks to the Author:

This study compares rotational correlation times and diffusion coefficient for single molecule with that for ensembles. A large decoupling is found between rotational and translational dynamics in both cases near T_g . These data establishes that the violation of Debye-Stokes-Einstein prediction is not the result of ensemble averaging. The breakdown is found to be a characteristic of the molecule itself.

It is also found that single molecule translations decouple from ensemble and appear slower as T_g is approached. Hence a degree of the breakdown may be attributable to ensemble averaging.

Analysis of single molecule trajectories shows that the breakdown is enhanced for molecules exhibiting a greater level of directional trajectories.

Moreover, molecules with greater radius of gyration were found to have greater diffusion coefficient and appear to drive the breakdown.

This work provides the first report of simultaneous measurement of rotational and translational dynamics in a molecular liquid, thereby unambiguously demonstrating breakdown of DSE at the molecular level. This work provides a valuable contribution to the field and may be published in Nature Communication.

We thank the reviewers for their thoughtful comments and reply below and via changes to the text to their specific questions and concerns.

We note specific changes below; overall, the most significant changes are to the Introduction and Reference sections; in the latter, new references 8-15, 37, 48-50, and 52-53 were added.

Reviewer #1 (Remarks to the Author):

The authors discuss translational and rotational molecular motions and their decoupling behavior of small solutes in polystyrene environments, and physical properties of the Stokes-Einstein-Debye relationship. The Stokes-Einstein (SE) and Stokes-Einstein-Debye (SED) relationship has been derived in hydrodynamic theorem on the basis of the Stokes equation, and their violations have been very highlighted in the large field of physical chemistry, applied physics, and materials chemistry. The authors employed experimental measurements of the fluorescent probe technique to quantify single-molecule and ensemble diffusions in polymer glass.

Their measured diffusion coefficients show the strong decoupling behavior of translational-rotational motions near the glass transition temperature, resulting the violation of SED relationship. The results of single-molecule and ensemble measurements agree well each other, and the single-molecule one is further applied to reveal structural effects of the gyration radius of diffusion solutes. I think that these experimental data help to understand nanoscale behavior and new physical insights on the SE and SED relationship in glassy polymer, and will attracts wide attentions of many readers in Nature Communications (in my opinion, Fig 3e is especially impressive). Thereby, I generally agree with the publication of this study from NatComm, but in my opinion, the following modifications should be required with careful consideration.

1.1 - in abstract: the authors said that translational diffusion coefficients nearly 400 times higher than expected from "SED", but I think that the translational diffusion should be related with the "SE", not SED. Why it's compared with the SED equations?

We were starting from the Stokes Einstein (SE) relationship,

$$D_T = \frac{kT}{6\pi\eta r_s}$$

We do not directly measure viscosity (nor are there good values for viscosity for this molecular weight of polystyrene in the relevant temperature range in the literature), so we cannot directly assess the validity of the SE relationship. Similarly, from our measurements, we cannot directly assess the validity of the Debye Stokes Einstein (DSE) relationship as expressed via,

$$\tau_c = \frac{4\pi\eta r_s^3}{3kT}$$

However, the product of D_T and τ_c , which is itself considered an alternative expression of the DSE relationship, can be assessed directly from our measurements.

$$D_T\tau_c = \frac{2}{9}r_s^2$$

This expression does not require knowledge of viscosity and predicts $D_T\tau_c$ is a constant and is independent of temperature. Because we do not measure viscosity, but do measure both D_T and τ_c , it is this combination of the SE and DSE relationships that we use as our basic guide to SE/DSE breakdown throughout the manuscript. This was discussed to a certain extent in Supporting 7 but we recognize that earlier introduction of these details is important. We have altered the Introduction to clarify these points.

1.2. - in abstract and main body: I'm confusing the phrase of "anomalous diffusion". The non-linear relaxation of MSD is well discussed like that, for example, in JCP 143, 191103 (2015). However, in this study, the time relaxation of MSD seems to be approximated as $MSD \sim 4Dt^1$, which is different with the above article. What is the correct mean of "anomalous diffusion" the authors said? (I think that this phrase would be sometimes misleading)

We thank the reviewer for this comment. Indeed, we had used "anomalous diffusion" in a manner that may have been confusing to readers. We agree that the term "anomalous diffusion" is most commonly associated with mean square displacements that do not grow linearly in time, showing either super-diffusive ($MSD \sim Dt^x$ with $x > 1$) or sub-diffusive ($MSD \sim Dt^x$ with $x < 1$) character. Somewhat surprisingly, we did find largely linear (Fickian) behavior of single molecule MSDs in time, consistent with simple Brownian diffusion. This was reinforced by our quasi-ensemble MSD (average of all single molecule MSDs, Fig. S4b), which also showed linear behavior and is less strongly affected by noise than are single molecule MSDs. We note that we do expect that if we had higher temporal and spatial resolution, we would find a region of sub-diffusion and evidence of caging behavior in these MSDs; we expect our spatial resolution would have to be < 1 nm to see this sub-diffusive regime associated with caging, and instead we are seeing the long time diffusive component of the MSDs. We note that a recent MD simulation study on supercooled water found similarly linear MSD vs. lag time plots, with linear behavior evident by 1 ns (and $MSD \sim 0.01$ nm²) for the lowest temperature probed even while the molecules experienced anomalously large translational jumps inconsistent with Brownian diffusion (Ref. 26 in the main text).¹

We also note that the term anomalous diffusion is sometimes used in cases in which MSDs appear linear. Most notably this was described in (new) Reference 37 in the main text, entitled "Anomalous yet Brownian."² As in that work, the MSDs we measured appeared linear, but we found evidence of behaviors inconsistent with simple Brownian diffusion. In particular, we found fat tails in step size distributions and a significant degree of anisotropy in the trajectories. We suspect that given the heterogeneity of dynamics in the system, linear MSDs may result from averaging over time for individual time lags as occurs in MSD analysis (in addition to due to the presence of noise). We are continuing to study such issues with the help of complementary simulations with tunable heterogeneity.

In sum, despite the largely linear MSDs, we do see non-Gaussian step size distributions and significant trajectory anisotropy that suggests simple Brownian diffusion is not occurring, which is also consistent with the strong degree of DSE breakdown found. To limit confusion, we have removed instances of the term "anomalous diffusion" in the text where we could find a suitable, compact replacement but have left it in other places, but now with additional comment and reference to the paper (Ref. 37) mentioned above.

1.3. - in the introduction of main body: The introduction of this manuscript is well written for the physical perspectives of dynamic heterogeneity, glass transition, and polymer chemistry.

However, the present story remains unkind for theoretical backgrounds of the SE and SED relationships with the Debye relaxation model. A lot of experimental and computational studies have been reported until now for applicability and its violation of the hydrodynamic relationships, which should be discussed in the introduction, and compared with the present focus for many readers in the wide varieties of scientific fields.

- lines 39, 40: the SE and SED relation should be emphasized to be derived on the basis of hydrodynamics. In the present case, the condition of small-molecule solute also does not follow the hydrodynamic assumption. However, for example, the following references empirically supports the applicability of SE and SED relationship under low-viscous condition, not high-viscous conditions. These references should be mentioned in the introduction.

-- J. Chem. Phys. 126, 224506 (2007).
-- J. Phys. Condens. Matter 33, 055401 (2021).

We have added discussion, citations (including those listed above, new References 11 and 12), and context to the Introduction regarding Stokes Einstein and Debye Stokes Einstein behavior and breakdown. We clarify that the SE and DSE relationships were developed for situations in which a large, spherical solute is present in a low viscosity solution, though the relationships have been found to hold in many additional cases, most notably for liquids with small tracer particles or for liquids in which self-diffusion is measured. The 2007 paper cited above has been one of significant interest to our group: it suggests that probes that are sufficiently small (by mass) report DSE breakdown in supercooled liquids while larger ones do not. We believe this is not because of fundamental assumptions of DSE associated with solute size, but because of the tendency for large probes to average over heterogeneous regions in these systems – if that occurs, any DSE breakdown that emerges causally from dynamic heterogeneity cannot be reported by the probe. The question of solute size compared to solvent size is also a bit difficult to assess in polymeric systems. Our previous work and that of others shows that rotational dynamics reported in studies of small fluorophore probes in polymers in the rubbery regime report segmental dynamics of the host, so likely the relevant “host” mass or volume is the portion of the polystyrene that constitutes a segment rather than the full chain. We thank the reviewer particularly for pointing out the 2021 paper as this was one we were not familiar with previously. We refer to it now in the manuscript as well as in response to Reviewer 2, Point 2.3.

1.4. - lines 43-45, page 1: the authors should clarify convincing citations for the sentence of "Most often, translational motion has been shown to exhibit a weaker temperature dependence than rotation, ". Furthermore, what is the mean of "Most often"? It's apparently related to the phrase of "anomalous diffusion"...

This sentence has now been made more precise and citations have been added. We meant to convey – and now state directly – that most experimental results quantifying rotational-translational decoupling have found a weaker temperature dependence of translation (diffusion coefficients) than rotation (rotational correlation times), resulting in a large enhancement of translation relative to rotation near T_g . This is not true in all systems – in particular, densely packed colloidal systems

or their analogs in simulation have sometimes shown DSE breakdown through unexpectedly slow rotation rather than enhanced translation relative to viscosity; similarly, supercooled water, which shows SE/DSE breakdown far above T_g , has shown a variety of anomalies, which may be related to particularities of the measurements or analysis and/or the liquid-liquid phase transitions present in this system.

1.5 - Fig 1: the bar plots of Fig.1, Fig.S1, Fig.3 etc is very unkind to identify the median and averaged value of single-molecule rotational correlation time from the Figures. The overlapped areas should be also shown for readers.

We have tried a variety of approaches to graphically depict median values on plots with distributions but have not found any approach that is fully satisfying. We have now added median values for Fig. 1 as dotted vertical lines in panels a and b; they were already represented (by points) in Fig. 1c. We have also added dotted vertical lines in Fig. S1 depicting the median values. For Fig. 3a, we do not ascribe any particular importance to the median values, instead concentrating on R_g values above the noise floor, with the noise floor shown via the dashed line. All molecules (regardless of temperature) with R_g to the right of that line are deemed mobile and to the left of that line immobile, with further description of those sets of molecules in the additional graphs associated with Fig. 3.

1.6 - Fig 2: If my understanding is correct, the product of D_T and τ_c is obtained from each state of single molecule observation. If so, the authors clarify the difference of notations for D_t and averaged (or median) D_T in the data of single molecule measurement; for example, Fig 2a uses the sampling set of (D_T^i, τ_c^i) and Fig 1c uses the averaged (or median) D_T and τ_c .

Thank you for pointing this out. We have now been explicit when we are discussing average, median, and/or single molecule reports of D_T and τ_c . This resulted in changes on pages 5, 7, and 8.

1.7 Furthermore, if it's possible, it's may be better to show the plots of Fig 2 as the contour map of probability density-of-states, because the most-probable state is hidden in the case of high-density scatter plot. Otherwise, please plot the median values of D_T and τ_c used in Fig 1c and discussed it.

We explored presenting this as a contour plot but believe that that obscures the fact that each point is from a single molecule. Instead, to enhance visualization, we have made each point significantly smaller. In addition, we have added median values on these graphs to facilitate comparison.

1.8 - Fig 2c: the authors discussed the effects of gyration radius in the SED relation, and concluded that the larger gyration radius enhances the SED violation. Meanwhile, the correlation of Fig 2c appear to be weak. Please change it to the probability density-of-state as above, or quantify the Pearson's product moment correlation coefficient. Furthermore, in main body, it's

better to discuss theoretical relationship with the equation of $D_T \tau_c$ for Fig 2c, as mentioned in Supporting Information.

When we state that larger radius of gyration correlates with greater breakdown of DSE, we mean that at every temperature the molecules with large R_g tend to have values furthest from the dashed line that is consistent with DSE behavior, $D_T \tau_c = \frac{2}{9} r_g^2$. We now state this explicitly. The Pearson's coefficient ranges between 0.46-0.64 across temperatures. While this value is not particularly high, it is higher than those seen in Fig. S8; for example, for R_g vs. τ_c across temperatures the maximum absolute value of the Pearson's coefficient is 0.17.

We have added discussion of the relationship $D_T \tau_c = \frac{2}{9} r_g^2$ to the Introduction and included this relationship in Fig. Caption 2.

1.9- Throughout the manuscript, the authors discuss the behavior of diffusion coefficients at the 4 states of different temperatures, but the solvent effects of polystyrene is not considered. The temperature difference changes the free-volume of small solutes, which affects the gyration radius of movable solutes plotted in Fig 2c. I think that the authors should mention this solvent effects of polymer glasses on translational-rotational decoupling and the SED relationship.

The polymer used in this study is heated under vacuum prior to measurements to ensure full removal of solvent, so solvent effects related to dissolving solvent (for either the polymer or probe) should not appear. We agree, however, that temperature likely changes the effective free volume around the probe molecules, as is captured by the probe rotational dynamics. Indeed, this fluorescent probe has been shown to report the temperature dependence of the segmental dynamics of the polymer previously and return the same temperature dependence as measured in probe-free or other-probe-bearing studies.^{3,4} As such, if the probe acts as a plasticizer or has any other intermolecular interactions of relevance in this study, it is not in a manner that is temperature-dependent and as such we do not believe it relevant in the observation of DSE violation or rotational-translational decoupling that increases as a function of decreasing temperature.

Reviewer #2 (Remarks to the Author):

2.1 In this manuscript, Mandel et al. showed single molecule experimental results of Debye-Stokes-Einstein (DSE) Breakdown in polystyrene glasses. In particular, the authors demonstrated translational-rotational decoupling towards the glass transition temperature by measuring the temperature dependence of the product of diffusion coefficient D and rotational correlation time τ_c . Furthermore, the degree of the DSE breakdown is explained by the mobile molecule contributions from the single molecular level analysis.

Although I'd appreciate the report of many results using single molecule experiments, the result of DSE breakdown is not surprising and a great deal in the present manuscript has already been addressed in previous papers by experiments and simulations, as cited by the authors. The

manuscript could be improved with more extensive discussions, particularly for what the new physical implication of DSE breakdown beyond current understanding is from single molecule analysis.

We have now attempted to highlight the novelty and importance of this study more fully in the manuscript. To start, as the reviewer appreciates, while DSE breakdown has been extensively documented in ensemble experiments, single molecule experiments tracking both translation and rotation simultaneously have not been performed before. This work is the first simultaneous single molecule study of rotation and translation in a system near T_g . This alone is of great import as an experimental advance critical to understanding DSE breakdown – specifically, we can conclude that rotational-translational decoupling is not solely a manifestation of different experimental approaches or analyses that unintentionally interrogate and report on a subset of molecules in the system. This had been one leading explanation for discrepancies in rotational measurements (such as dielectric spectroscopy), which are posited to report preferentially on particularly slow molecules, and translational measurements (for example, NMR studies sensitive to translation), which are posited to preferentially weight fast molecules. Single molecule experiments provide the unique ability to remove potential sub-ensemble selection and/or weighting. If this proposed explanation had been the sole cause of observed DSE breakdown, each individual molecule would have shown a diffusion coefficient consistent with its rotational correlation time as predicted through DSE. *This manuscript provides the first experimental evidence of a large degree of DSE breakdown within single molecules, showing that ensemble level averaging cannot explain the majority of DSE breakdown and quantifying the degree that **can** be explained by ensemble averaging.* The study then allows further interrogation of the potential origins of the portion of the breakdown that is a single molecule phenomenon in which individual molecules exhibit translational mobility orders of magnitude faster than that which would be expected from their rotational mobility. The work suggests that individual molecule averaging over heterogeneous dynamics and/or coupled structural and dynamic domains (i.e. islands of slow molecules separated by “rivers” of fast molecules) are present in these systems. While some of these findings are similar to ones that have been revealed through or could be studied via simulations, we note that even with significant advances, simulations still do not typically probe supercooled systems near T_g , as such simulations would need to cover hundreds of seconds to probe temperatures where dynamic heterogeneity and its sequelae become obvious.

Beyond this, several additional advances are present in this study. 1) This study provides the first direct comparison between single molecule and ensemble (fluorescence correlation imaging) measurements using the same probe in the same host system, allowing *direct comparison* of degree of breakdown that emerges from an ensemble technique to that present at the single molecule level. 2) The study provides unprecedented localization accuracy for single molecule measurements in a condensed phase system. Of the few single molecule translational studies that exist in such systems, the lowest diffusion coefficient reported was $\approx 30 \text{ nm}^2/\text{s}$ whereas that presented here is $< 1 \text{ nm}^2/\text{s}$, which can only be accessed due to the localization accuracy of $< 10 \text{ nm}$ achieved in this study^{5,6}. Such high resolution provides access to details of molecular motion as they occur on length scales suggested by simulations to be the ones relevant to spatial heterogeneity.

Below, I'd like to list several questions and comments.

2.2 The definition of SE/DSE relationships should be discussed in a more strict manner. Indeed, there have been other presentations of SE/DSE relationships, including $\eta/(\tau_c T)$ with viscosity η and D/D_r with rotational diffusion constant D_r , different from $D\tau_c$. Why $D\tau_c$ only? If those quantities would be assessed and compared consistently, that would be beyond the current understanding of SE/DSE breakdowns in glassy systems.

Thank you for this comment. Several points on this matter are discussed in response to questions and comments of Reviewer 1. In particular, Points 1.1 and 1.3 above discuss these questions. In response to these comments and those of Reviewer 1, we have deepened our discussion of these points in the Introduction. In response to the particular question above, viscosity data is very challenging to obtain in supercooled liquids. We quote the following, from a recent paper on supercooled water: “However, viscosity data are needed for a direct test of SE and SED relations. Quite surprisingly, there are only two sets of data for the viscosity η at significant supercooling. ... However, the two sets disagree.”⁷ This was similar to the situation we found in the literature with regard to viscosity measurements of high molecular weight polystyrene in the rubbery regime. We note that even in simulations, assessing viscosity is not straightforward. A recent paper noted, “In a computer simulation, the self-diffusion coefficient is easily found from the long-time mean-square displacement, whereas determining the (shear) viscosity is much more involved.”⁸

Given the challenges in assessing viscosity in the supercooled regime, where it changes enormously with small changes in temperature, the behavior of $D_T\tau_c$ has been commonly used as a measure of experimental SE/DSE breakdown, and this quantity was relatively straightforward to access through our measurements. We are very interested also in comparing $D_T\tau_c$, which is also theoretically accessible in our experiments and is the focus of ongoing study. However, as pointed out in Ref. 9, “The use of the rotational diffusion constant D_r needs particular care due to the limitation of the angular Brownian motion scenario. Furthermore, it has been revealed that D_r is superficial for describing the reorientational motion in supercooled molecular liquids.”⁹ These challenges limit our discussion of D_r at this time.

In sum, the DSE relationship captured through $D_T\tau_c = \frac{2}{9}r_s^2$ provided a relatively straightforward approach that allowed direct comparison between single molecule and ensemble studies presented here as well as ensemble measurements presented in the past.

2.3 In the Debye model, the rotational correlation depends on the order of spherical harmonics. It is important to discuss the order dependence of τ_c because the validation of the Debye model is the basis of DSE quantity, $D\tau_c$.

We have read closely the work we mention in Point 2.2 above on the spurious breakdown of DSE⁹ and how this may relate to experiments that probe distinct spherical harmonics in determining rotational correlation time. Reference 10 also discusses how τ dependence on temperature may vary as a function of spherical harmonics. Linear dichroism correlation functions are not actually related to a single spherical harmonic but probe a timescale close to τ_2 (new References 48-50).¹¹⁻¹³ While we recognize that there can be some difference in temperature dependence of τ_c depending on the experimental observation (and spherical harmonic) from which it is derived, the differences

seen in these cases amount to factors of typically no more than 3. We see a difference in the value of $D_T\tau_c$ as a function of temperature that varies by 2-3 orders of magnitude and cannot be attributed to small variations that may emerge from differences in particular rotational correlation time assessed. We now point to this explicitly in the manuscript on pages 3 and 4.

2.4 The term "decoupling" is used many times in the manuscript, but it seems ambiguous. In fact, Figure 2(a) shows a correlation between D and τ_c , indicating that more diffusion leads to more rotation. It may be "translational-rotational coupling", but would be wrong?

Decoupling, here, is used to denote the fact that rotations slow much more quickly than translations as temperature is decreased, as can be appreciated most directly from Fig. 1c. This is the same usage as in the titles of References 21, 26, 30, and 31 in the main text. It is true that in our work as temperature decreases, diffusion coefficient becomes smaller and rotational correlation time becomes larger. Additionally, it is true that on average faster rotating molecules tend to be faster translating molecules, though this correlation as seen in Fig. 2 is far from perfect (Pearson's R across temperatures = -0.58 but within any given temperature is maximally 0.17, with some temperatures showing weak positive correlation and other temperatures showing weak negative correlation between these quantities). Despite these correlations, on average, molecular rotational correlation time changes much more over the temperature window explored than does diffusion coefficient, inconsistent with DSE behavior. Indeed, as the temperature is decreased, molecules are found to exhibit a greater amount of translational motion per rotation, which would be a strikingly unusual finding in any system. We have clarified our use of the term rotational-translation decoupling in the Introduction on page 2.

2.5 The contribution of mobile molecules to DSE breakdown is interesting. It is also interesting to investigate the effects of immobile counterparts on DSE relationship. From Figure 2(b), it would be expected that $D*\tau_c$ of the immobile subset will exhibit an almost recovery of DSE relation. If so and if not so, what is the implication of immobile molecules in DSE breakdowns?

This is an interesting point and others have pursued similar questions in simulations (for example, Reference 26 in the main text).¹ That study found that excluding particularly mobile molecules in supercooled water resulted in a system that obeyed DSE predictions, though we note that this was a case where the violation was found to be no more than a factor of 3-4, not orders of magnitude as we measure in this study and others have measured in polystyrene previously.

The set of molecules that we deem "immobile" do not reflect DSE behavior in our measurements – this can be seen in Fig. 2a and b, with very few molecules sitting near or on the line associated with DSE behavior. It is true that they do exhibit reduced breakdown relative to mobile molecules and compared to all molecules considered together. It may also be true that some (or most or even all) of the remaining deviation may be an apparent deviation rather than a real deviation from DSE behavior. As defined in our study, immobile molecules exhibit mobility that we cannot distinguish from error, so the calculated diffusion coefficients of these molecules may be dominated by error. Indeed, many molecules identified as immobile return negative (unphysical) diffusion coefficients, and thus their contribution to DSE breakdown cannot be readily assessed. Enhancing spatial localization through increasing the number of photons collected and enhancing the particle

tracking aspect of our data analysis will allow better characterization of the behavior of the slowest molecules in the ensemble. We are continuing to pursue these goals.

2.6 Another interesting quantify to highlight between mobile and immobile molecules is the ratio of $\langle \tau_c \rangle \langle 1/\tau_c \rangle$ or $\langle D \rangle \langle 1/D \rangle$. Is it possible to measure those values?

These values are certainly of interest given the idea that $\langle D_T \rangle$ (and $\langle 1/\tau_c \rangle$) will highlight fast contributions and $\langle \tau_c \rangle$ (and $\langle 1/D_T \rangle$) will highlight slow ones in an ensemble. Examining $\langle \tau_c \rangle \langle 1/\tau_c \rangle$ and $\langle D_T \rangle \langle 1/D_T \rangle$ from our measurements reveals nearly temperature independent values of ≈ 2 for the former and ≈ 8 for the latter. This is consistent with our findings of nearly temperature independent widths of the τ_c and D_T distributions and the fact that the D_T distributions are wider than those found for τ_c in our measurement. We are continuing to think through the consequences of this finding as well as the various formulations of the SE and DSE relationships.

Reviewer #3 (Remarks to the Author):

This study compares rotational correlation times and diffusion coefficient for single molecule with that for ensembles. A large decoupling is found between rotational and translational dynamics in both cases near T_g . These data establishes that the violation of Debye-Stokes-Einstein prediction is not the result of ensemble averaging. The breakdown is found to be a characteristic of the molecule itself.

It is also found that single molecule translations decouple from ensemble and appear slower as T_g is approached. Hence a degree of the breakdown may be attributable to ensemble averaging. Analysis of single molecule trajectories shows that the breakdown is enhanced for molecules exhibiting a greater level of directional trajectories. Moreover, molecules with greater radius of gyration were found to have greater diffusion coefficient and appear to drive the breakdown. This work provides the first report of simultaneous measurement of rotational and translational dynamics in a molecular liquid, thereby unambiguously demonstrating breakdown of DSE at the molecular level. This work provides a valuable contribution to the field and may be published in Nature Communication.

We thank the reviewer for their positive review.

References

1. Dueby, S., Dubey, V. & Daschakraborty, S. Decoupling of Translational Diffusion from the Viscosity of Supercooled Water: Role of Translational Jump Diffusion. *J. Phys. Chem. B* **123**, 7178–7189 (2019).
2. Wang, B., Anthony, S. M., Bae, S. C. & Granick, S. Anomalous yet Brownian. *Proc. Natl. Acad. Sci. U.S.A.* **106**, 15160–15164 (2009).
3. Paeng, K. & Kaufman, L. J. Single Molecule Experiments Reveal the Dynamic Heterogeneity and Exchange Time Scales of Polystyrene near the Glass Transition. *Macromolecules* **49**, 2876–2885 (2016).
4. Manz, A. S., Paeng, K. & Kaufman, L. J. Single molecule studies reveal temperature independence of lifetime of dynamic heterogeneity in polystyrene. *J. Chem. Phys.* **148**, 204508 (2018).
5. Flier, B. M. I. *et al.* Heterogeneous diffusion in thin polymer films as observed by high-temperature single-molecule fluorescence microscopy. *J. Am. Chem. Soc.* **134**, 480–488 (2012).
6. Flier, B. M. I. *et al.* Single molecule fluorescence microscopy investigations on heterogeneity of translational diffusion in thin polymer films. *Phys. Chem. Chem. Phys.* **13**, 1770–1775 (2011).
7. Dehaoui, A., Issenmann, B. & Caupin, F. Viscosity of deeply supercooled water and its coupling to molecular diffusion. *Proc. Natl. Acad. Sci. U.S.A.* **112**, 12020–12025 (2015).
8. Costigliola, L., Heyes, D. M., Schröder, T. B. & Dyre, J. C. Revisiting the Stokes-Einstein relation without a hydrodynamic diameter. *J. Chem. Phys.* **150**, 021101 (2019).
9. Kawasaki, T. & Kim, K. Spurious violation of the Stokes–Einstein–Debye relation in supercooled water. *Sci. Rep.* **9**, 8118 (2019).
10. Singh, J. & Jose, P. P. Violation of Stokes–Einstein and Stokes–Einstein–Debye relations in polymers at the gas-supercooled liquid coexistence. *J. Phys. Condens. Matter* **33**, 055401 (2021).
11. Mackowiak, S. A., Herman, T. K. & Kaufman, L. J. Spatial and temporal heterogeneity in supercooled glycerol: Evidence from wide field single molecule imaging. *J. Chem. Phys.* **131**, 244513 (2009).
12. Wei, C.-Y. J., Kim, Y. H., Darst, R. K., Rossky, P. J. & Vanden Bout, D. A. Origins of Nonexponential Decay in Single Molecule Measurements of Rotational Dynamics. *Phys. Rev. Lett.* **95**, 173001 (2005).
13. Vallée, R. A. L. *et al.* Analysis of the exponential character of single molecule rotational correlation functions for large and small fluorescence collection angles. *J. Chem. Phys.* **128**, 154515 (2008).

Reviewers' Comments:

Reviewer #1:

Remarks to the Author:

The authors have properly improved the manuscript as following the comments in the first round review. The background and important references of SE and SED relations are clearly mentioned in the main text, and the discussed properties are well described with the theoretical equations (for example, as shown in the equation of " $D_T \tau_c = (2/9) r_s^2$ "). The revised discussions are meaningful for understanding the SE and SED relationships, and Figure 2 is also improved to clearly show molecular perspectives of SED equation for diluted molecular solutes in glassy polymers. I think that many readers in NatComm can benefit from the revised manuscript.

Therefore, in my opinion, this article is of enough quality for the publication from Nature Communications.

Reviewer #2:

Remarks to the Author:

The authors responded to comments of my first-round review. The reply is appropriate and I'd now recommend the paper will be publishable in Nature Communications. However, one more modification must be necessary for the main text before the publication.

With regard to the definition of DSE relation (point 2.2), the discussion of the rotational correlation time τ_c and the rotational diffusion constant D_r is still missing in the introduction.

The authors' reply to point 2.2 can improve the main text. Also, the paper by Chong and Kob [Phys. Rev. Lett. 102, 025702 (2009)] should be cited in the literature.

Reviewer #3:

Remarks to the Author:

The authors have satisfactorily answered the reviewers questions and the manuscript is now ready for publication.

We thank the reviewers for their comments and respond below explicitly to Reviewer #2 who had some additional comments and recommendations.

“With regard to the definition of DSE relation (point 2.2), the discussion of the rotational correlation time τ_c and the rotational diffusion constant D_r is still missing in the introduction.”

“The authors' reply to point 2.2 can improve the main text. Also, the paper by Chong and Kob [Phys. Rev. Lett. 102, 025702 (2009)] should be cited in the literature.”

We had previously included the following in Response 2.2 (see earlier response for references within):

“In response to the particular question above, viscosity data is very challenging to obtain in supercooled liquids. We quote the following, from a recent paper on supercooled water: “However, viscosity data are needed for a direct test of SE and SED relations. Quite surprisingly, there are only two sets of data for the viscosity η at significant supercooling. ... However, the two sets disagree.”⁷ This was similar to the situation we found in the literature with regard to viscosity measurements of high molecular weight polystyrene in the rubbery regime. We note that even in simulations, assessing viscosity is not straightforward. A recent paper noted, “In a computer simulation, the self-diffusion coefficient is easily found from the long-time mean-square displacement, whereas determining the (shear) viscosity is much more involved.”⁸

Given the challenges in assessing viscosity in the supercooled regime, where it changes enormously with small changes in temperature, the behavior of $D_{T\tau_c}$ has been commonly used as a measure of experimental SE/DSE breakdown, and this quantity was relatively straightforward to access through our measurements. We are very interested also in comparing $D_{T\tau_c}$, which is also theoretically accessible in our experiments and is the focus of ongoing study. However, as pointed out in Ref. 9, “The use of the rotational diffusion constant D_r needs particular care due to the limitation of the angular Brownian motion scenario. Furthermore, it has been revealed that D_r is superficial for describing the reorientational motion in supercooled molecular liquids.”⁹ These challenges limit our discussion of D_r at this time.

In sum, the DSE relationship captured through $D_{T\tau_c} = \frac{2}{9} r_s^2$ provided a relatively straightforward approach that allowed direct comparison between single molecule and ensemble studies presented here as well as ensemble measurements presented in the past.”

In the introduction, we now refer explicitly to the formulation of the DSE with D_r and also include reference to the Chong and Kob paper (new reference 16) – this is a paper we were familiar with and it was an oversight not to cite it, as it clearly describes why D_r is not the most useful quantity with which to assess DSE breakdown.